# Elevated serum YKL-40, IL-6, CRP, CEA, and CA19-9 combined as a prognostic biomarker panel after resection of colorectal liver metastases

Reetta Peltonen[1]☯*, Mathias H. Gramkow[2]☯, Christian Dehlendorff[3], Pia J. Osterlund[4,5]‡, Julia S. Johansen[2,6,7]‡, Helena Isoniemi[1]‡

1 Transplantation and Liver Surgery, Abdominal Center, University of Helsinki and Helsinki University Hospital, Helsinki, Finland, 2 Department of Oncology, Herlev and Gentofte Hospital, Copenhagen University Hospital, Herlev, Denmark, 3 Statistics and Pharmacoepidemiology, Danish Cancer Society Research Center, Danish Cancer Society, Copenhagen O, Denmark, 4 Department of Oncology, Tampere University Hospital, and University of Tampere, Tampere, Finland, 5 Helsinki University Hospital, Department of Oncology, and University of Helsinki, Helsinki, Finland, 6 Department of Medicine, Herlev and Gentofte Hospital, Copenhagen University Hospital, Herlev, Denmark, 7 Department of Clinical Medicine, Faculty of Health and Medical Sciences, University of Copenhagen, Copenhagen K, Denmark

☯ These authors contributed equally to this work.
‡ These authors also contributed equally to this work.
* reetta.peltonen@hus.fi

**Data Availability Statement:** All relevant data are within the manuscript and its Supporting Information files.

## Abstract

### Background

The inflammatory biomarkers, YKL-40 and interleukin-6 (IL-6), are elevated in patients with metastatic colorectal cancer. We examined their associations with relapse-free survival and overall survival in combination with serum C-reactive protein (CRP), carcinoembryonic antigen (CEA), and carbohydrate antigen 19–9 (CA19-9) in patients with colorectal liver metastases.

### Methods

Altogether 441 consecutive patients undergoing liver resection at Helsinki University Hospital between 1998 and 2013 were included in the study. Pre- and postoperative YKL-40 and IL-6 were determined from serum samples with commercially available enzyme-linked immunosorbent assay (ELISA) kits, and CRP, CEA, and CA19-9 by routine methods. Associations between these biomarkers and relapse-free and overall survival were examined using Cox regression analysis.

### Results

Patients with 2–5 elevated biomarkers were at an increased risk of relapse compared to those with 0–1 elevated biomarkers, preoperatively (HR 1.37, 95% CI 1.1–1.72) or postoperatively (HR 1.54, 95% CI 1.23–1.92). Patients with 2–5 elevated biomarkers were also at

**Funding:** This study was financially supported by the Competitive State Research Financing of the Expert Responsibility Area of Helsinki University Hospital (RP, PO, HI) and Tampere University Hospital (PO), the Cancer Foundation Finland (RP, PO, HI), Suomen Onkologiayhdistys (RP), the Danish Cancer Society (MG), and Finska Läkaresällskapet (PO). The YKL-40 and IL-6 kits and analysis were funded by the Department of Oncology, Herlev and Gentofte Hospital, Copenhagen University Hospital, Herlev, Denmark. The funders had no role in study design, data collection and analysis, decision to publish, or preparation of the manuscript.

**Competing interests:** The authors have declared that no competing interests exist.

an increased risk of death compared to those with 0–1 elevated biomarkers, preoperatively (HR 1.76, 95% CI 1.39–2.24) or postoperatively (HR 1.83, 95% CI 1.44–2.33).

## Conclusion

The results suggest that a protein panel of the inflammatory biomarkers YKL-40, IL-6, and CRP, and the cancer biomarkers CEA and CA19-9 might identify patients that benefit from more aggressive treatment and surveillance, although the additional value of IL-6 and CRP in this aspect is limited.

## Introduction

In 2018, more than 860 000 patients died from colorectal cancer (CRC) worldwide [1]. A large number, up to 75%, of patients with CRC develop colorectal liver metastases (CRLM) [2]. If the metastases are confined to the liver and other resectable extrahepatic sites, resection with curative attempt is recommended, if technically and physiologically feasible [3]. Five- and ten-year survival rates after liver resection have been in the range of 38% and 26%, respectively, but a significant number of the resected experience an early recurrence [3]. As relapses are frequent and median overall survival (OS) short (median 3.6 years in a review), prognostic scoring systems have been developed for identifying patients who might benefit from intensified adjuvant therapy and follow-up, combined with a possible re-resection upon recurrence [3–5]. A well-defined, universally accepted adjuvant therapy and surveillance regimen after liver resection is yet to be established for this patient group.

Serum carcinoembryonic antigen (CEA) is widely used as a biomarker for detection of recurrent CRC and for monitoring the response to systemic therapy [6], but its prognostic value leaves room for improvement and added arsenal, as around 20% of CRC tumors are CEA negative [7]. Carbohydrate antigen 19–9 (CA19-9), also called sialyl-Lewis$^A$, is a tetrasaccharide carbohydrate synthesized by the gastrointestinal epithelium. Several studies have shown that elevated CA19-9 levels were strong negative prognostic factors in different stages of CRC [8, 9]. Finding new biomarkers in addition to CEA and CA19-9 would help estimate the risk of recurrence and determine a suitable adjuvant therapy and follow-up protocol for patients with metastatic CRC (mCRC) after liver resection.

Tumor-promoting inflammation is an enabling characteristic of cancer and enhanced systemic inflammatory response has been linked to impaired survival [10, 11]. Inflammatory biomarkers are of special interest in the search for new valuable prognostic biomarkers on top of established pathologic and demographic factors in patients resected for CRLM. Serum C-reactive protein (CRP), YKL-40 (also called chitinase-3-like-1 protein, CHI3L1) and interleukin-6 (IL-6) are established biomarkers of inflammation [9, 12, 13]. In a recent study, serum C-reactive protein (CRP), the production of which is stimulated by IL-6 [14], was found to be significantly associated with shorter OS in patients undergoing liver resection [15]. IL-6 and YKL-40 are secreted by cancer cells and macrophages [12, 16], and the latter also by neutrophils [12]. YKL-40 could play an important role in promoting tumor metastasis and impairing survival via multiple mechanisms, such as upregulating the secretion of matrix metalloproteinases, activating the TGFβ pathway [17], inducing angiogenesis via its receptor IL-13α2 and membrane protein TMEM219, and binding to RAGE (receptor for advanced glycation end products) [12]. IL-6 stimulates the production of YKL-40 and angiogenesis via its receptors IL-6R and sIL-6R [12, 18–20]. Elevated serum YKL-40 and IL-6 levels are seen in patients with diagnosed

CRC or those later developing gastrointestinal cancers [13, 21], and higher levels are linked to impaired survival [12, 15].

The early detection of cancer with emerging technologies, such as liquid biopsies that enable the detection of circulating tumor cells, cell-free DNA (cfDNA) or circulating tumor DNA (ctDNA) with specific mutations in blood or other body fluids, is under active investigation [22]. However, protein biomarkers are at present an integral part of clinical work and may be more easily accessed.

In this retrospective biomarker study, we examined the prognostic value of preoperative and postoperative serum YKL-40, IL-6, CRP, CEA, and CA19-9 in a cohort of 441 consecutive patients who had undergone liver resection for CRLM. Our hypothesis was that the patients with combined high serum concentrations of YKL-40, IL-6, CRP, CEA, and CA19-9 would have the worst relapse-free survival (RFS) and OS in patients with mCRC treated with curative-intent liver resection.

## Materials and methods

### Patients

Altogether 455 consecutive patients undergoing liver resection for mCRC were enrolled in this study between March 1998 and February 2013 at the Helsinki University Hospital, Helsinki, Finland (S1 Fig). All patients included were diagnosed with resectable CRLM, and those with possible other malignancies were excluded. Liver metastases were defined as synchronous, if they were diagnosed at the same time or within 6 months after the operation of the primary tumor, and as metachronous, if diagnosed later. The liver resections were considered major if more than two Couinaud segments were resected. Extrahepatic disease was excluded with whole-body contrast-enhanced computed tomography (CT) and in some cases with positron emission tomography (PET). Neoadjuvant and/or adjuvant treatment were given according to guidelines at the time of surgery. Fourteen patients were excluded from the statistical analyses due to the following reasons: 7 patients had extrahepatic metastases that eventually could not be surgically removed, in 6 cases a radical liver resection could not be performed, and 1 patient died within 20 days after surgery, and thus, the prognostic value of the serum markers could not be evaluated.

Serum samples were taken before liver resection and approximately 3 months after operation. Median time from preoperative blood sampling to surgery was 16 days (interquartile range (IQR) 9–29 days). Median time from surgery to postoperative sampling was 94 days (IQR 89–98 days).

The study was conducted in accordance with the Declaration of Helsinki, and it was approved by the Ethics Committee at Helsinki University Hospital (IRB99/07/01, HUS531/E6/01, HUS460/E6/05, HUS323/13/3/2008, HUS242/13/03/02/2011). Collection and analysis of blood samples were approved by the National Supervisory Authority for Welfare and Health (Valvira) of Finland (STM Dno 4858/04/047/08), and information about the dates of death was obtained from the Central Statistical Office of Finland (TK-53-1004-9). A verbal informed consent was obtained from all the patients included in the study, and the collection of the research samples signified consent. The collection of blood samples was initiated over twenty years ago, and written consent was not required at that time according to the Finnish law. The REMARK guidelines have been used in reporting [23].

### Serum samples

The pre- and postoperative venous blood samples were collected in gel tubes. The samples were centrifuged within 30–120 minutes, and serum was stored at -80˚C. YKL-40 and IL-6

were determined in duplicate in both pre- and postoperative serum samples using commercially available enzyme-linked immunosorbent assays (ELISAs); YKL-40: MicroVue YKL-40 ELISA (Catalog #8020), Quidel Corporation, San Diego, CA, USA; and IL-6: Quantikine HS600B, R&D Systems, Abingdon, OX, UK; according to the manufacturer's instructions. For YKL-40, the detection limit was 20 ng/ml, and intra- and inter-assay coefficients of variation (CVs) <5% and <6% [24]. For IL-6, the detection limit was 0.01 pg/ml, and intra- and inter-assay CVs ≤8% and ≤11% [25]. The pre- and postoperative serum levels of CRP, CEA and CA19-9 were determined with automatic analyzers as follows: CRP: immunoturbidimetric method (1998–2013) at Huslab laboratories, Helsinki University Hospital; CEA and CA19-9: immunoenzymatic assay, Bayer Immuno 1 (CEA: 1998–10/2005; and CA19-9: 1998–1/2006), or immunochemiluminometric assay, Abbott Architect (CEA: 10/2005–2013; and CA19-9: 1/2006–2013). All measurements were performed by technicians blinded to the study endpoints.

An age-corrected 95th percentile for serum YKL-40 in healthy individuals served as the cut-off level for elevated values, and it was calculated using the formula proposed by Bojesen et al. [24]:

$$P_{i\%} = \frac{100}{(1 + YKL - 40^{-3} \cdot (1.062^{Age}) \cdot 5000}$$

The cut-off values for the other biomarkers were the following: 4.95 pg/ml (P95% for serum IL-6 in healthy individuals [25]); 5 mg/l for CRP; 5 µg/l for CEA; and 37 kU/l for CA19-9.

## Statistical analyses

Wilcoxon signed-rank test was used for estimating differences between pre- and postoperative serum values and Spearman's rank test for testing correlations. For OS, the time to event was defined as time from the date of liver surgery to the date of death from any cause or censored at the end of follow-up, and for RFS as time from surgery to recurrence or censored at the end of follow-up. The cut-off date for follow-up was October 31st, 2017.

Serum YKL-40, IL-6, CRP, CEA, and CA19-9 were included in the Cox regression analyses as log2-transformed continuous variables due to non-normality and as elevated or normal values according to the above-mentioned cut-off levels in the Kaplan-Meier plots. Biomarker values below the limit of detection (LOD) were imputed as LOD/2 as a pragmatic approach, and serum values of zero seemed unlikely. Crude and adjusted hazard ratios (HR) with 95% confidence interval (CI) were estimated for RFS and OS using Cox regression. Survival curves were estimated using the Kaplan-Meier method and equality of strata was tested for by a likelihood ratio test. The variables included in the multivariate analyses (Tables 3 and 4) were the following: serum values of YKL-40, IL-6, CRP, CEA, CA19-9, age, gender, location of the primary tumor, presentation of liver metastases (synchronous/metachronous), type of liver resection (minor/major), the number and size of the liver metastases, and the resection margins (R0/1/2).

Proportional hazards assumption was evaluated by testing for trends in the scaled Schoenfeld residuals. A cut point analysis was carried out for all investigated markers with regards to their discriminatory power for 3-year mortality and 3-year recurrence using receiver operating characteristic (ROC) curves and calculating the area under the curve (AUC). The cut point with a sensitivity closest to 80% was chosen. All statistical analyses were carried out using the statistical software R version 3.3.3 (R Core Team, Vienna, Austria). A p-value of less than 0.05 was considered significant, and only two-sided tests were used.

## Results

Patient characteristics and the pre- and postoperative serum concentrations of YKL-40, IL-6, CRP, CEA, and CA19-9 are shown in Tables 1 and 2, respectively. The median age of the patients was 64.9 years, and 59% of them were men. During the follow-up period, 265 patients (60%) experienced a recurrence, of which 242 (91%) were diagnosed within 3 years after liver resection. The median RFS was 1.7 years (IQR: 0.6–6.6 years). Median OS was 5.8 years (IQR: 5.4–8.4 years), and 268 patients (61%) died during follow-up. Median follow-up time was 5.7 years.

The results of the cut point analysis showed that only elevated postoperative CEA had an AUC larger than 0.7 ($AUC_{CEA}$ = 0.73, 95% CI 0.66–0.79) for predicting mortality within 3 years (S1 Table and S2 Fig). Based on the cut point analyses of the markers, we chose to use the pre-specified cut-offs as the optimal cut points did not differ markedly from them (see S1 Table–S4 Table and S1 Fig for results of the cut point analyses).

The characteristics of the pre- and postoperative biomarker values are shown in Table 2. YKL-40 and IL-6 increased after liver resection: the median increase in YKL-40 levels was 13 ng/ml (range -1261–571, p = 0.015) and in IL-6 levels 1.7 pg/ml (range -19.4–73.9, p<0.001).

**Table 1. Baseline patient characteristics and outcome data.**

| Variable | Overall (N = 441) |
|---|---|
| Sex | |
| Male | 260 (59.0%) |
| Female | 181 (41.0%) |
| Age in years, median (min.–max.) | 64.9 (33–84) |
| Location of the primary tumor | |
| Right colon | 84 (19.0%) |
| Left colon | 174 (39.5%) |
| Rectum | 183 (41.5%) |
| Presentation of liver metastases[1] | |
| Synchronous | 254 (57.6%) |
| Metachronous | 187 (42.4%) |
| Number of liver metastases, median (max.) | 1 (16) |
| Size of the largest metastasis, median (max.) | 2.5 cm (15.0) |
| Type of liver resection[2]: | |
| Minor | 201 (45.6%) |
| Major | 239 (54.2%) |
| Missing | 1 |
| Resection margins | |
| R0 | 412 (93.6%) |
| R1 | 19 (4.3%) |
| R2 | 9 (2.0%) |
| OS after liver resection, median (min.–max.) | 5.8 years (0.0–19.7) |
| RFS after liver resection, median (min.–max.) | 1.7 years (0.04–19.3) |
| Alive at the end of follow-up | 172 (39.0%) |
| Alive without recurrence | 123 (27.9%) |
| Recurrence or death | 318 (72.1%) |

[1] Diagnosed within 6 months after the operation of the primary tumor (synchronous) or later (metachronous).

[2] 1–2 (minor) or ≥3 (major) Couinaud segments resected.

**Table 2. Pre- and postoperative biomarker concentrations.**

| Biomarker | Preoperative | Postoperative |
|---|---|---|
| YKL-40 (number) | 413 | 413 |
| Median (min.–max.) (ng/ml) | 74 (20–1504) | 87 (20–744) |
| Elevated (>age-corrected 95th percentile), n (%) | 57 (13.8) | 78 (18.9) |
| IL-6 (number) | 413 | 413 |
| Median (min.–max.) (pg/ml) | 3.4 (0.3–33.5) | 5.1 (0.6–80.7) |
| Elevated (>4.95 pg/ml), n (%) | 142 (34.4) | 214 (51.8) |
| CRP (number) | 429 | 434 |
| Median (min.–max.) (mg/l) | <5 (<5–149) | <5 (<5–153) |
| Elevated (>5 mg/l), n (%) | 86 (20.0) | 129 (29.7) |
| CEA (number) | 440 | 440 |
| Median (min.–max.) (µg/l) | 5.2 (0.5–853) | 2.5 (0.5–294) |
| Elevated (>5 µg/l), n (%) | 221 (50.2) | 74 (16.8) |
| CA19-9 (number) | 437 | 436 |
| Median (min.–max.) (kU/l) | 13 (<2–11 310) | 9 (<2–33 800) |
| Elevated (>37 kU/l), n (%) | 99 (22.7) | 43 (9.9) |

Preoperative YKL-40 correlated with preoperative IL-6 (rho = 0.39, p<0.001) and preoperative CEA (rho = 0.11, p = 0.024), and postoperative YKL-40 with postoperative IL-6 (rho = 0.28, p<0.001) and postoperative CEA (rho = 0.15, p = 0.002). Preoperative CRP correlated with preoperative IL-6 (rho = 0.31, p<0.001), YKL-40 (rho = 0.21, p<0.001) and CEA (rho = 0.25, p<0.001). Preoperative CEA correlated with preoperative CA19-9 (rho = 0.41, p<0.001) and CRP (rho = 0.25, p<0.001), and postoperative CEA with postoperative CA19-9 (rho = 0.22, p<0.001). Postoperative CA19-9 correlated with postoperative IL-6 (rho = 0.13, p = 0.008). Age correlated with pre- and postoperative YKL-40 (rho = 0.25, p<0.001; and rho = 0.33, p<0.001, respectively), with preoperative IL-6 (rho = 0.16, p = 0.001), and with postoperative CA19-9 (rho = 0.11, p = 0.027). There was no significant correlation between age and CEA.

## Association of relapse-free survival with YKL-40, IL-6, CRP, CEA, and CA19-9

The results of the univariate and multivariate analyses for RFS are shown in Table 3. All the coefficients in the Cox regression analyses for pre- and postoperative variables in regard to RFS and OS can be found in the S5 Table.

Higher preoperative $\log_2$-transformed YKL-40 (HR 1.19), IL-6 (HR 1.15) and CA19-9 (HR 1.08) were associated with shorter RFS in the univariate analysis, and CA19-9 in the multivariate analysis (HR 1.12). Patients with 3 elevated biomarkers concurrently were at an increased risk of relapse compared to those with no elevated biomarkers in the univariate (HR 1.60; Fig 1A) and in the multivariate analyses (HR 1.63). Patients with 2–5 elevated markers compared to those with 0–1 had shorter RFS in univariate (HR 1.37; Fig 2A) and in multivariate (HR 1.40) analyses.

To investigate whether YKL-40, IL-6 and/or CRP drove this association, an explorative multivariate analysis was carried out. It showed that neither preoperatively elevated CEA and/or CA19-9 nor elevated YKL-40, IL-6 and/or CRP associated with shorter RFS (S6 Table).

Higher postoperative $\log_2$-transformed YKL-40 (HR 1.21), IL-6 (HR 1.11), CEA (HR 1.24), and CA19-9 (HR 1.12) all were associated with shorter RFS in the univariate analyses, but CRP

**Table 3. Results of the Cox regression analyses for relapse-free survival.**

| | Univariate | p-value | Multivariate[1] | p-value |
|---|---|---|---|---|
| | HR (95% CI) | | HR (95% CI) | |
| **Preoperatively measured biomarkers** | | | | |
| YKL-40 | 1.19 (1.07–1.32) | <0.001 | 1.08 (0.96–1.22) | 0.212 |
| IL-6 | 1.15 (1.03–1.28) | 0.010 | 1.04 (0.91–1.18) | 0.566 |
| CRP | 1.08 (0.96–1.21) | 0.219 | 1.03 (0.88–1.20) | 0.756 |
| CEA | 1.01 (0.96–1.07) | 0.583 | 1.00 (0.93–1.07) | 0.952 |
| CA19-9 | 1.08 (1.03–1.14) | <0.001 | 1.12 (1.06–1.19) | <0.001 |
| 0 elevated (N = 111) | Reference | | Reference | |
| 1 elevated (N = 113) | 0.81 (0.58–1.12) | 0.202 | 0.81 (0.58–1.12) | 0.202 |
| 2 elevated (N = 101) | 1.13 (0.83–1.56) | 0.438 | 1.21 (0.87–1.68) | 0.252 |
| 3 elevated (N = 44) | 1.60 (1.08–2.36) | 0.018 | 1.63 (1.08–2.47) | 0.021 |
| 4 elevated (N = 29) | 1.57 (0.99–2.49) | 0.055 | 1.88 (1.12–3.16) | 0.016 |
| 5 elevated (N = 1) | 2.86 (0.40–20.7) | 0.298 | 3.60 (0.48–26.8) | 0.211 |
| 0–1 elevated (N = 224) | Reference | | Reference | |
| 2–5 elevated (N = 175) | 1.37 (1.10–1.72) | 0.005 | 1.40 (1.11–1.78) | 0.005 |
| **Postoperatively measured biomarkers** | | | | |
| YKL-40 | 1.21 (1.09–1.34) | <0.001 | 1.06 (0.94–1.19) | 0.323 |
| IL-6 | 1.11 (1.01–1.23) | 0.033 | 0.98 (0.87–1.11) | 0.755 |
| CRP | 1.07 (0.99–1.15) | 0.075 | 1.02 (0.92–1.12) | 0.741 |
| CEA | 1.24 (1.14–1.34) | <0.001 | 1.17 (1.07–1.29) | <0.001 |
| CA19-9 | 1.12 (1.04–1.20) | 0.002 | 1.09 (1.01–1.17) | 0.029 |
| 0 elevated (N = 137) | Reference | | Reference | |
| 1 elevated (N = 99) | 1.18 (0.87–1.62) | 0.290 | 1.00 (0.73–1.38) | 0.993 |
| 2 elevated (N = 106) | 1.41 (1.05–1.91) | 0.023 | 1.30 (0.95–1.78) | 0.102 |
| 3 elevated (N = 48) | 1.98 (1.37–2.86) | <0.001 | 1.84 (1.25–2.71) | 0.002 |
| 4 elevated (N = 10) | 2.94 (1.48–5.86) | 0.002 | 2.50 (1.22–5.12) | 0.012 |
| 5 elevated (N = 3) | 3.67 (1.16–11.7) | 0.028 | 3.14 (0.97–10.1) | 0.055 |
| 0–1 elevated (N = 236) | Reference | | Reference | |
| 2–5 elevated (N = 167) | 1.54 (1.23–1.92) | <0.001 | 1.57 (1.24–1.98) | <0.001 |

[1] Variables included in the multivariate analyses: serum values of IL-6, YKL-40, CRP, CEA, and CA19-9; age; gender; location of the primary tumor; type of liver metastases (synchronous/metachronous; Table 1); type of liver resection (minor/major; Table 1); the number and size of the liver metastases; and the resection margins (R0/1/2). The biomarkers are presented as $\log_2$-transformed continuous variables.

was not. In the multivariate analysis, the association between shorter RFS and higher CEA and CA19-9 remained statistically significant (Table 3).

The patients with 2–5 elevated biomarkers postoperatively were at an increased risk of relapse compared to those with 0–1 elevated (HR 1.54; Figs 3A and 4A), and this association remained statistically significant in the multivariate analysis (HR 1.57). Compared with zero elevated biomarkers, having CEA and/or CA19-9 elevated was significantly associated with shorter RFS, while elevated YKL-40, IL-6 and/or CA19-9 were not (S6 Table).

## Association of overall survival with YKL-40, IL-6, CRP, CEA, and CA19-9

The results of the univariate and multivariate analyses are shown in Table 4.

Higher preoperative $\log_2$-transformed YKL-40 (HR 1.29), IL-6 (HR 1.16), CEA (HR 1.07), and CA19-9 (HR 1.12) were associated with shorter OS in the univariate analyses, but CRP

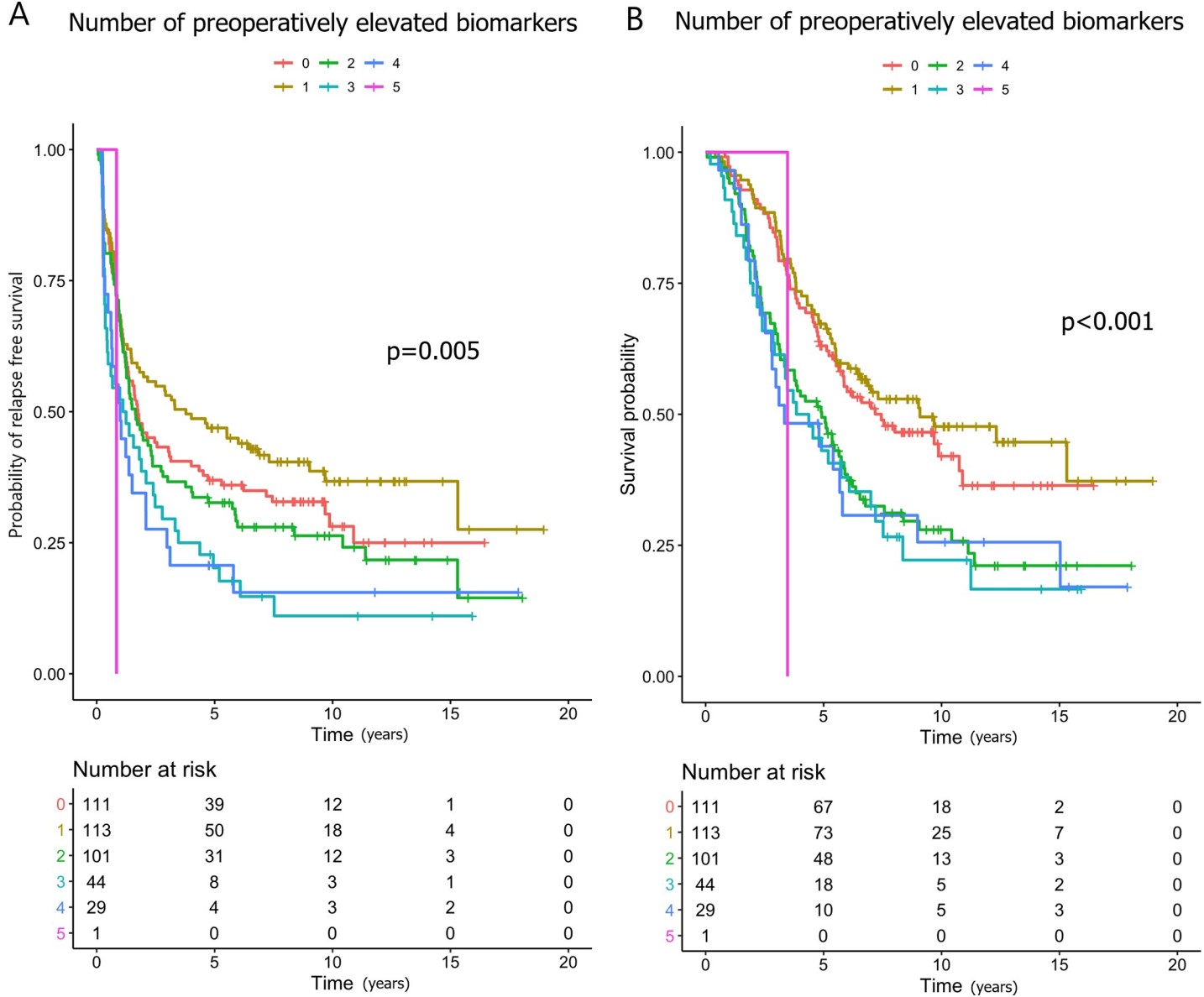

**Fig 1.** Kaplan-Meier curves showing the associations between the numbers of elevated preoperative biomarkers and (A) relapse-free survival and (B) overall survival.

was not. Higher YKL-40 and CA19-9 remained significant in the multivariate analysis (HR 1.19, and HR 1.13, respectively).

The patients with 2–5 elevated biomarkers preoperatively were at an increased risk of death compared to those with 0–1 elevated biomarkers (HR 1.76; Figs 1B and 2B), and this association remained statistically significant in the multivariate analysis (HR 1.71). Neither preoperatively elevated CEA and/or CA19-9 nor elevated YKL-40, IL-6 and/or CRP associated significantly with shorter OS, when taken in combination (S6 Table).

Higher postoperative $\log_2$-transformed values of all biomarkers associated with shorter OS in the univariate analyses (YKL-40: HR 1.26; IL-6: HR 1.13; CRP: HR 1.11; CEA: HR 1.33; and CA19-9: HR 1.22), and CEA (HR 1.24) and CA19-9 (HR 1.17) remained significant in the multivariate analysis (Table 4).

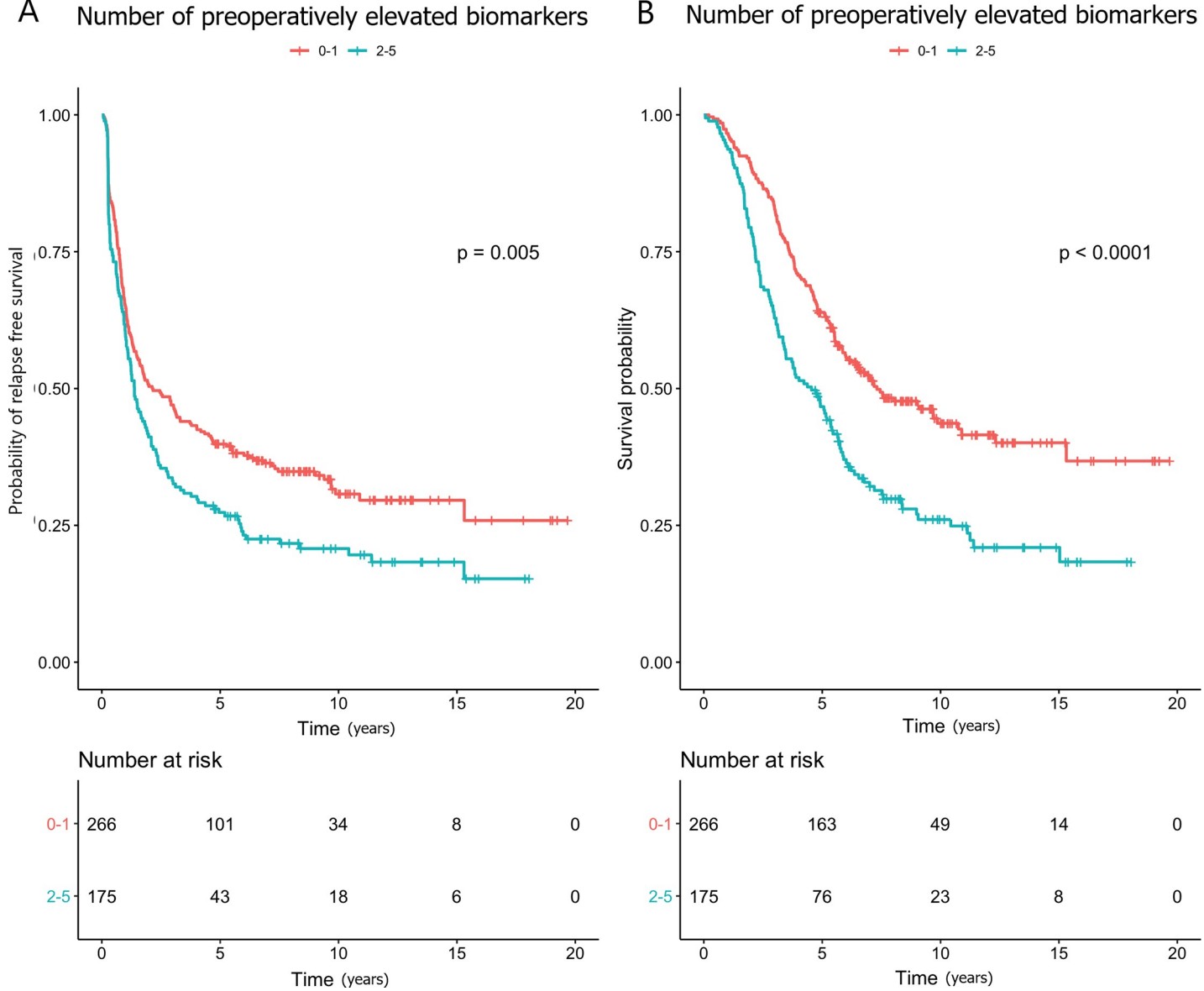

**Fig 2.** Kaplan-Meier curves showing the associations between 0–1 and 2–5 elevated preoperative biomarkers and (A) relapse-free survival and (B) overall survival.

The patients with 2–5 elevated biomarkers postoperatively were at an increased risk of death compared to those with 0–1 elevated (HR: 1.83; Figs 3B and 4B). In the multivariate analysis, this association remained significant (HR 1.84). Compared with zero elevated biomarkers, having CEA and/or CA19-9 elevated was significantly associated with shorter OS, while elevated YKL-40, IL-6 and/or CA19-9 were not (S6 Table).

## Discussion

In this biomarker study of patients with colorectal liver metastases, we found that higher serum values of YKL-40 and CA19-9 preoperatively were associated with shorter OS. Similar results were found for postoperatively higher serum values of CEA and CA19-9

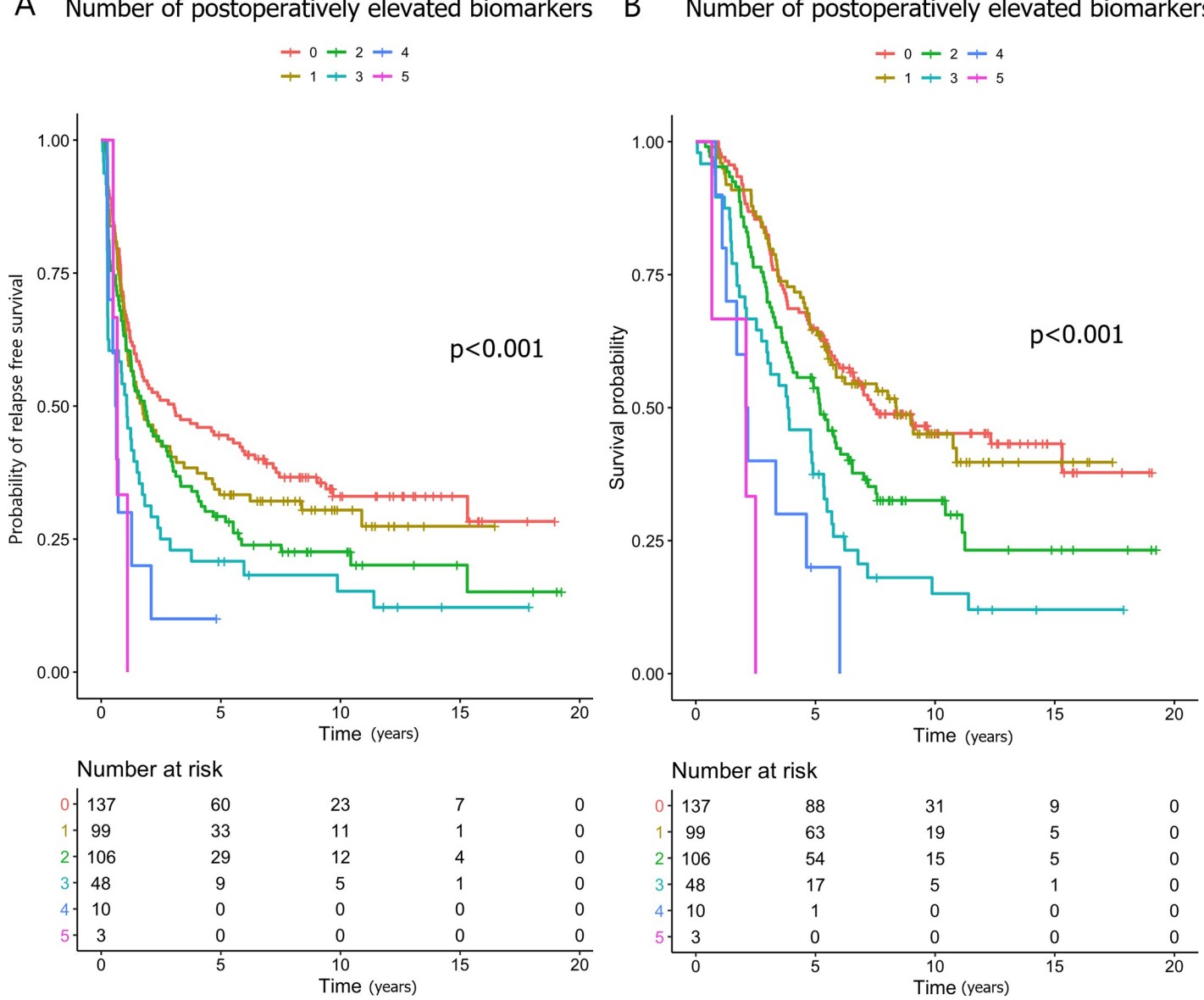

**Fig 3.** Kaplan-Meier curves showing the associations between the numbers of elevated postoperative biomarkers and (A) relapse-free survival and (B) overall survival.

postoperatively. Interestingly, we found that patients with two or more elevated biomarkers pre- or postoperatively had a shorter OS, suggesting that a combined panel of inflammatory biomarkers could be used in combination with the tumor biomarkers CEA and CA19-9. Higher serum values of YKL-40 and CA19-9 preoperatively and higher serum values of CEA and CA19-9 postoperatively associated with shorter RFS, and a combination of all the biomarkers showed that patients with two or more elevated biomarkers postoperatively had a shorter RFS. No previous studies have evaluated the combined prognostic value of these 5 biomarkers.

Our results are in line with recent studies examining the prognostic value of serum YKL-40 [26], CEA [8, 9, 27, 28], and CA19-9 [8, 9, 27] as biomarkers in patients with mCRC. CRP has recently been found to have prognostic value in patients with mCRC, although our results

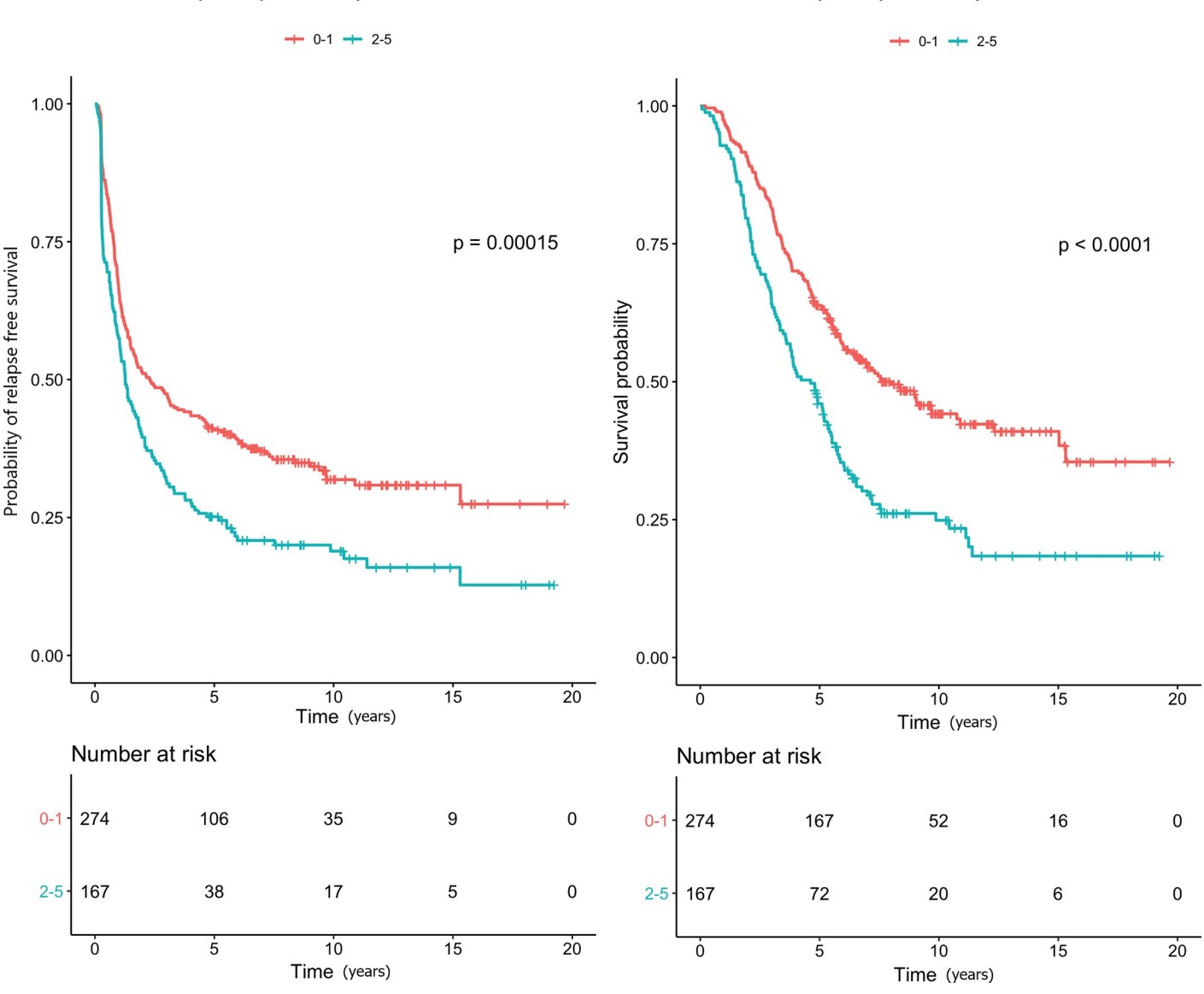

**Fig 4.** Kaplan-Meier curves showing the associations between 0–2 and 2–5 elevated postoperative biomarkers and (A) relapse-free survival and (B) overall survival.

do not directly support this finding [29]. The association between serum IL-6 and OS in patients with CRC is unclear, since some previous studies have found higher IL-6 levels to be significantly associated with shorter OS, while others have failed to show any association [16, 29, 30]. We did not find a direct association between IL-6 and RFS or OS, when adjusted for other biomarkers.

YKL-40 has recently been found to be involved in the formation of cancer metastases through regulation of epithelial to mesenchymal transformation and subsequent migration/invasion [31]. In addition, YKL-40 has been shown to regulate VEGF in tumor cells and to promote angiogenesis, enabling tumors to spread via a hematological pathway [12]. This may explain why we see high serum YKL-40 levels in patients with mCRC and, furthermore, why high serum YKL-40 seems to reflect relapse after liver resection. In a recent study it was shown

**Table 4. Results of the Cox regression analyses in relation to overall survival.**

| | Univariate | p-value | Multivariate[1] | p-value |
|---|---|---|---|---|
| | HR (95% CI) | | HR (95% CI) | |
| **Preoperatively measured biomarkers** | | | | |
| YKL-40 | 1.29 (1.16–1.44) | <0.001 | 1.19 (1.04–1.35) | 0.010 |
| IL-6 | 1.16 (1.03–1.30) | 0.011 | 1.03 (0.90–1.18) | 0.685 |
| CRP | 1.07 (0.95–1.21) | 0.240 | 0.91 (0.77–1.06) | 0.215 |
| CEA | 1.07 (1.01–1.12) | 0.019 | 1.02 (0.95–1.09) | 0.602 |
| CA19-9 | 1.12 (1.06–1.18) | <0.001 | 1.13 (1.06–1.20) | <0.001 |
| 0 elevated (N = 111) | Reference | | Reference | |
| 1 elevated (N = 113) | 0.86 (0.60–1.24) | 0.433 | 0.82 (0.57–1.18) | 0.284 |
| 2 elevated (N = 101) | 1.63 (1.16–2.30) | 0.005 | 1.58 (1.11–2.24) | 0.011 |
| 3 elevated (N = 44) | 1.82 (1.19–2.79) | 0.006 | 1.60 (1.02–2.50) | 0.042 |
| 4 elevated (N = 29) | 1.78 (1.08–2.93) | 0.023 | 1.75 (1.01–3.04) | 0.046 |
| 5 elevated (N = 1) | 3.24 (0.45–23.5) | 0.245 | 2.90 (0.39–21.7) | 0.299 |
| 0–1 elevated (N = 224) | Reference | | Reference | |
| 2–5 elevated (N = 175) | 1.76 (1.39–2.24) | <0.001 | 1.71 (1.33–2.21) | <0.001 |
| **Postoperatively measured biomarkers** | | | | |
| YKL-40 | 1.26 (1.13–1.41) | <0.001 | 1.10 (0.97–1.24) | 0.131 |
| IL-6 | 1.13 (1.02–1.26) | 0.022 | 0.95 (0.83–1.09) | 0.498 |
| CRP | 1.11 (1.02–1.20) | 0.011 | 1.09 (0.99–1.20) | 0.096 |
| CEA | 1.33 (1.22–1.46) | <0.001 | 1.24 (1.12–1.36) | <0.001 |
| CA19-9 | 1.22 (1.13–1.31) | <0.001 | 1.17 (1.08–1.28) | <0.001 |
| 0 elevated (N = 137) | Reference | | Reference | |
| 1 elevated (N = 99) | 1.03 (0.72–1.47) | 0.882 | 0.88 (0.61–1.27) | 0.488 |
| 2 elevated (N = 106) | 1.57 (1.13–2.17) | 0.007 | 1.40 (0.99–1.97) | 0.055 |
| 3 elevated (N = 48) | 2.42 (1.64–3.56) | <0.001 | 2.33 (1.56–3.48) | <0.001 |
| 4 elevated (N = 10) | 4.45 (2.21–8.96) | <0.001 | 4.42 (2.16–9.05) | <0.001 |
| 5 elevated (N = 3) | 10.2 (3.16–32.9) | <0.001 | 8.91 (2.71–29.2) | <0.001 |
| 0–1 elevated (N = 236) | Reference | | Reference | |
| 2–5 elevated (N = 167) | 1.83 (1.44–2.33) | <0.001 | 1.84 (1.43–2.37) | <0.001 |

[1] Variables included in the multivariate analyses: serum values of IL-6, YKL-40, CRP, CEA, and CA19-9; age; gender; location of the primary tumor; type of liver metastases (synchronous/metachronous; Table 1); type of liver resection (minor/major; Table 1); the number and size of the liver metastases; and the resection margins (R0/1/2). The biomarkers are presented as $\log_2$-transformed continuous variables.

that paracrine signaling of IL-6 in synergy with IL-8 directly promoted cell migration and is thus important in the formation of metastasis [32]. Recently, IL-6 has also been found to be involved in the formation of a metastatic niche in the liver [33], which indicates that it could be associated with a negative outcome in CRC. The symptoms of inflammation, i.e. sarcopenia and cachexia, that accompany many cancers may be related to higher IL-6 levels in serum, and they are presently being investigated relative to treatment efficacy also in mCRC [34]. One option that has been suggested is treatment with the monoclonal antibody tocilizumab, which targets the IL-6 receptor, given in combination with standard chemotherapy in other indications [35, 36].

The reason for investigating IL-6, CRP and YKL-40 in combination, in spite of the fact that IL-6 primarily stimulates YKL-40 and the production of CRP in the liver [14], is that YKL-40 is also stimulated by other biological processes and proinflammatory cytokines [12]. In addition, we can assume that the production of CRP is affected by liver metastases. This makes

studying all biomarkers in our patient cohort interesting, as the normal pathways may be interrupted by the pathophysiology of the metastases. Our results show that while there are clear associations between the studied inflammatory biomarkers, YKL-40 in particular seems to be singularly associated with RFS, which cannot be accounted for by IL-6. Several studies have shown an association between high serum YKL-40 and IL-6 and shorter OS in different types of cancers and in patients with severe inflammatory diseases [12, 37, 38], which limits their specificity. In addition, serum YKL-40 has been shown to be increased until the 21st postoperative day merely due to the surgical trauma related to primary tumor resection [39]. We sampled blood a median of 94 (IQR 89–98) days after surgery. Thus, the postoperative increase seen in serum YKL-40 and IL-6 in some of the patients could be caused by micro-metastatic disease rather than the large surgical trauma in the liver, although it is possible that some effect of the surgical trauma persisted. We assume that the postoperative changes in inflammatory biomarkers would have normalized before the postoperative sampling.

Elevated postoperative CEA was associated with shorter RFS and OS, as has also been found in previous studies [9, 28]. A systematic review by Spelt et al. [40] showed a large variation in cut-off values and in the prognostic value of CEA measured before liver resection for CRLM, which naturally leads to the conclusion that different cut-off values may influence the results. We used a cut-off value of 5 μg/l since higher cut-offs have no impact on sensitivity in our cut-point analysis.

High preoperative and postoperative CA19-9 associated with poor prognosis, which is in line with previous studies [8, 9, 27, 28]. Despite these findings, the role of adding CA19-9 to CEA in the follow-up of CRC patients has not yet been established. Our results strengthen the importance of CA19-9 as a prognostic and predictive marker in mCRC.

In our explorative analysis, further investigating the observed enhanced risk of relapse or death when having an increased number of elevated markers, it seemed that CA19-9 and CEA were the main drivers of the association. These biomarkers were also more consistently associated with a negative outcome, when analyzed on a continuous scale. We acknowledge that the use of 2–5 elevated biomarkers is non-specific, but it suggests which combinations of markers most reliably predict the outcome of patients. The additive value of the inflammatory markers to a standard biomarker may thus be limited, but they could identify selected patients, since the risk of relapse or death increases with each additional elevated marker. Further research will be needed to elucidate this aspect, and we are aware that the added value of IL-6 and CRP is limited in the present study.

It is probable that new technologies, liquid biopsies in particular, will be applied to clinical use quite shortly. They have potential in the detection and management of CRC, as they offer a non-invasive method for early detection of cancer, prognostic and predictive information, monitoring of treatment response, and identification of minimal residual disease [41]. In conjunction with resection of CRLM, elevated pre- [42] and postoperative [43] ctDNA levels have shown high concordance with relapse in ctDNA-positive patients. However, it seems unlikely that liquid biopsies could provide information on inflammatory responses, which affect the prognosis of CRC. Thus, the inflammatory biomarkers have an important role also in this era of advances in these new technologies.

A limitation of our study is the lack of a validation cohort. We did not consider the diurnal variation of IL-6 [44], which may confound our results. It would also be interesting to study the value of YKL-40, IL-6, CRP, CEA and CA19-9 in consecutive blood samples after surgery to see whether repeated measurements give more information on the patients' prognoses and lead-time to relapse. We did not adjust for multiple testing by for example Bonferroni-Holm correction of p-values. This represents a drawback of the study and limits the confidence in our findings. The strengths of our study are the fairly large number of consecutive patients

included, the long follow-up time, and the reliable follow-up data with no patients lost to follow-up. Information about the *KRAS*, *NRAS* and *BRAF* mutational status as well as the microsatellite instability (MSI) status could have been valuable, since the mutational status has been shown to predict recurrence patterns after liver resection [45, 46], but the mutational status was not routinely analyzed at Helsinki University Hospital prior to 2013 and the MSI not prior to 2018.

## Conclusions

Pre- and postoperative serum levels of a panel of three inflammatory biomarkers YKL-40, IL-6, and CRP with the two cancer biomarkers CEA and CA19-9 showed that the patients with 2 or more elevated biomarkers had shorter RFS and OS after resection of liver metastases. In the future, this panel might be used to select the patients that could benefit from more aggressive perioperative chemotherapy and follow-up, although the role of IL-6 and CRP needs to be explored further.

## Supporting information

**S1 Fig. Study flow diagram.**
(DOC)

**S2 Fig. ROC curves describing true positive (TP) and false positive (FP) rates for predicting mortality within 3 years from baseline for a random subpopulation of 25% of the entire cohort.**
(DOCX)

**S1 Table. AUC and sensitivity analysis for predicting relapse-free survival 3 years after liver resection for pre-specified cut-off values as defined in the legend.**
(DOCX)

**S2 Table. Cut point analysis for all biomarkers investigated.** The cut-off was optimized as the one closest to providing a sensitivity of 80% for predicting progression-free survival 3 years after liver resection.
(DOCX)

**S3 Table. AUC and sensitivity analysis for predicting mortality 3 years after liver resection for pre-specified cut-off values as defined in the legend.**
(DOCX)

**S4 Table. Cut point analysis for all biomarkers investigated.** The cut-off was optimized as the one closest to providing a sensitivity of 80% for predicting mortality 3 years after liver resection.
(DOCX)

**S5 Table. (**A-D) Cox regression analysis estimates for all variables for both pre- and postoperative biomarker values and associations with (A and B) overall survival and (C and D) relapse-free survival.
(DOCX)

**S6 Table. Explorative Cox regression analysis of combinations of elevated markers.**
(DOCX)

**S1 File.**
(XLSX)

## Acknowledgments

We thank Noora Ask (Department of Transplantation and Liver Surgery, Abdominal Center, University of Helsinki and Helsinki University Hospital, Helsinki, Finland) for her valuable help with collecting the previously stored serum samples, and Ulla Kjærulff-Hansen and Marianne Sørensen (Department of Medicine, Herlev and Gentofte Hospital, Copenhagen University Hospital, Denmark) and Mie Barthold Krüger (Department of Oncology, Herlev and Gentofte Hospital, Copenhagen University Hospital, Denmark) for excellent technical assistance with the YKL-40 and IL-6 ELISA measurements.

## Author Contributions

**Conceptualization:** Pia J. Osterlund, Julia S. Johansen, Helena Isoniemi.

**Data curation:** Reetta Peltonen, Julia S. Johansen, Helena Isoniemi.

**Formal analysis:** Christian Dehlendorff.

**Funding acquisition:** Pia J. Osterlund, Julia S. Johansen, Helena Isoniemi.

**Investigation:** Reetta Peltonen, Mathias H. Gramkow.

**Methodology:** Reetta Peltonen, Mathias H. Gramkow, Christian Dehlendorff, Pia J. Osterlund, Julia S. Johansen, Helena Isoniemi.

**Project administration:** Julia S. Johansen, Helena Isoniemi.

**Resources:** Pia J. Osterlund, Julia S. Johansen, Helena Isoniemi.

**Software:** Christian Dehlendorff.

**Supervision:** Julia S. Johansen, Helena Isoniemi.

**Validation:** Christian Dehlendorff.

**Visualization:** Reetta Peltonen, Mathias H. Gramkow, Christian Dehlendorff, Pia J. Osterlund, Helena Isoniemi.

**Writing – original draft:** Reetta Peltonen, Mathias H. Gramkow.

**Writing – review & editing:** Reetta Peltonen, Mathias H. Gramkow, Christian Dehlendorff, Pia J. Osterlund, Julia S. Johansen, Helena Isoniemi.

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
