## [Decision Letter · Decision Letter 0]

15 Jan 2020

PONE-D-19-31995

Elevated serum YKL-40, IL-6, CRP, CEA, and CA19-9 combined as a prognostic biomarker panel after resection of colorectal liver metastases

PLOS ONE

Dear Dr. Peltonen,

Thank you for submitting your manuscript to PLOS ONE. After careful consideration, we feel that it has merit but does not fully meet PLOS ONE’s publication criteria as it currently stands. Therefore, we invite you to submit a revised version of the manuscript that addresses the points raised during the review process.

We would appreciate receiving your revised manuscript by Feb 29 2020 11:59PM. To enhance the reproducibility of your results, we recommend that if applicable you deposit your laboratory protocols in protocols.io, where a protocol can be assigned its own identifier (DOI) such that it can be cited independently in the future. For instructions see: http://journals.plos.org/plosone/s/submission-guidelines#loc-laboratory-protocols

We look forward to receiving your revised manuscript.

Kind regards,

Eugene J. Koay, M.D., Ph.D.

Academic Editor

PLOS ONE

Journal Requirements:

2. Please provide additional details regarding participant consent. In the ethics statement in the Methods and online submission information, please ensure that you have specified whether consent was written or verbal/oral. If consent was verbal/oral, please specify: 1) whether the ethics committee approved the verbal/oral consent procedure, 2) why written consent could not be obtained, and 3) how verbal/oral consent was recorded. If your study included minors, please state whether you obtained consent from parents or guardians in these cases. If the need for consent was waived by the ethics committee, please include this information.

Additional Editor Comments (if provided):

Reviewers' comments:

Reviewer's Responses to Questions

**Comments to the Author**

1. Is the manuscript technically sound, and do the data support the conclusions?

Reviewer #1: No

Reviewer #2: Yes

2. Has the statistical analysis been performed appropriately and rigorously? 

Reviewer #1: Yes

Reviewer #2: Yes

3. Have the authors made all data underlying the findings in their manuscript fully available?

Reviewer #1: Yes

Reviewer #2: No

4. Is the manuscript presented in an intelligible fashion and written in standard English?

Reviewer #1: Yes

Reviewer #2: Yes

5. Review Comments to the Author

Reviewer #1: The authors here present data from an impressively large number of patients who underwent resection of liver metastases for their colorectal cancer. They examine the role of 5 biomarkers and conclude elevated expression of a combination of these biomarkers associate with worse survival outcomes. These findings are consistent with prior efforts and fit into the general understanding of clinical management of metastatic colorectal cancer. I think that the message could be strengthened with additional points to be addressed:

1. None of the markers for systemic inflammation are associated with clinical outcomes as a single marker in the multivariate analysis, and the significance is seen only when analyzed "in combination." It is unclear which markers, when "2 or more" were elevated, were actually elevated. If one is to translate these to the clinic, should he or she order all five, or were there specific ones which seemed to be more associated with outcome? As presented, the applicability here is very difficult for the reader.

2. The authors mention that these findings support the use of these markers as a predictive biomarker and "might identify patients who benefit from more adjuvant chemotherapy." However, while there analyses may support the combination of markers PROGNOSTICALLY, the authors provide zero detail of receipt of any adjuvant chemotherapy for these patients. Therefore these claims are unsupported and should be removed as presented from the manuscript.

3. One could argue that if this combination of markers is associated with a poor prognosis, the biology of the given CRC tumor is especially poor, and these might identify the patients who undergo resection that will have a higher risk of relapse and therefore NOT benefit from more surgery if at especially high risk for relapse and shortened OS. I disagree that re-resection would be warranted based on the data presented.

4. If IL-6 is responsible for producing CRP and YKL-40, then if IL-6 is elevated, should the other downstream markers be as well? If so, what is the reason for needed to test all three? IL-6 alone did not identify high-risk patients in the multivariate survival analysis. The authors could consider addressing this in their discussion.

5. Clarification of "major" and "minor" hepatic resections (how are these defined?) is needed in the methods section.

6. YKL-40, IL-6 both increased from the preoperative to postoperative setting. One would expect that if these were associated with cancer only, there would be some decrease in median levels following resection, unless most patients recurred in the time before the first postoperative blood specimen was drawn. How do the authors interpret the fact that %patients with elevated inflammatory markers rose after surgery? Could these nonspecific markers be reflective of other factors?

7. Without knowing which markers were elevated (2 of 5, 3 of 5 -which ones???), the clinical applicability of these findings is low, especially since, as the authors mention, these markers can be attributed to other cancers, autoimmune diseases, etc. The cost effectiveness of this approach is unclear, and the omission of alternate emerging approaches as a prognostic/predictive biomarker such as ctDNA in this setting by the authors is glaring. I am not sure the conclusions from the authors are supported by their data.

Reviewer #2: I would like the commend the authors on a extremely well written manuscript which seeks to evaluate whether elevated serum biomarker levels of YKL-40, IL-6, and CRP may improve upon prognostic significance beyond CEA and CA 19-9. The authors perform a retrospective evaluation of biomarker study of 441 consecutive patients undergoing resection for colorectal liver metastases, where the collected samples prospectively preoperatively and postoperatively. With a median follow-up of 5+ years, the authors found that patients with 2-5 biomarkers elevated were at increased risk of relapse and worse OS compared to those with 0-1 biomarkers elevated. This is a large cohort of patients with a well defined population of patients, with potential for important prognostic and therapeutic implications of treatment. I would ask the authors to clarify the following points to further strengthen their manuscript:

1) Given the awareness and prognostic significance of mutational status, as the authors acknowledged, this manuscript would be further improved if mutational status were available to correlate with inflammatory biomarkers. Was microsatellite status available on any of these patients?

2) From a statistical standpoint, was Bonferroni correction applied to adjust for multiple testing, given the authors evaluation of outcomes categorized as: 0 elevated/1 elevated/ 2 elevated/3 elevated/ 4 elevated/5 elevated and then subsequently as: 0-1 elevated/2-5 elevated. Also, were other categorizations evaluated as well (i.e 0-2 elevated/3-5 elevated) and if so, it further strengthens the role for application of Bonferroni correction.

3) How was 3 months postop chosen as the optimal time point for this analysis? Was there subsequent serial draws at other time points as part of the study as well?

4) Given increasing data evaluating early detection of disease recurrence with liquid biopsies, the authors should mention it as emerging technologies that are challenging the prognostic significance of traditional biomarkers like CEA/CA 19-9.

5) Interesting that CA 19-9 was the only significant biomarker on MVA in the preoperative setting for RFS and only CEA and CA 19-9 were significant in the postoperative setting on MVA. This suggests that perhaps CEA and CA 19-9 (known prognostic factors) are what is driving the significance of the 2-5 biomarkers, and not YKL-40, IL-6 and CRP. Where any of the other clinical variables on MVA significant?

6) Table 1 as current presented is confusing and hard to follow. I would suggest that the authors remove outcome data from the Table 1 (alive/dead), and then present the biomarker data in the more concise and readable manner. Also please add R0/R1/R2 rates to Table. 1

7) On page 10, line 201: Please edit to include "WERE".....Ca 19-9 (HR 1.08) WERE associated

8) Similalry, on page 12: Please edit to include "WERE"....CA 19-9 (HR 1.12) WERE associated...

6. PLOS authors have the option to publish the peer review history of their article (what does this mean?). If published, this will include your full peer review and any attached files.

Reviewer #1: No

Reviewer #2: No

---

## [Author Response · Author response to Decision Letter 0]

28 Feb 2020

Responses to the comments of the Academic Editor and the Reviewers

We thank the Academic Editor and the Reviewers for thorough and accurate comments and constructive criticism. We have revised the manuscript according to the suggestions and responded to the comments point by point (please see below). We strongly feel that these changes have improved the manuscript.

Journal Requirements:

- We have edited our manuscript, where necessary, to ensure that it meets the style requirements.

2. Please provide additional details regarding participant consent. In the ethics statement in the Methods and online submission information, please ensure that you have specified whether consent was written or verbal/oral. If consent was verbal/oral, please specify: 1) whether the ethics committee approved the verbal/oral consent procedure, 2) why written consent could not be obtained, and 3) how verbal/oral consent was recorded. If your study included minors, please state whether you obtained consent from parents or guardians in these cases. If the need for consent was waived by the ethics committee, please include this information.

- We have added these details to our manuscript and online submission information as follows:

1) The Ethics Committee at Helsinki University Hospital has approved the verbal consent procedure.

2) The collection of blood samples was initiated over twenty years ago, and at that time, verbal consent was generally accepted. Many of the patients included in our study have already died, and the permission to use the samples in recent studies has been given by the National Supervisory Authority for Welfare and Health (Valvira) of Finland.

3) The verbal consent was taken by the treating physician(s) and passed on to the nurse in charge of ordering all the necessary laboratory specimens. The collection of the research samples signified participant consent.

4) No minors were included in this study.

- We have now uploaded a fully anonymized data set as a Supporting Information file, as requested.

Reviewers' comments:

Reviewer's Responses to Questions

Comments to the Author

1. Is the manuscript technically sound, and do the data support the conclusions?

Reviewer #1: No

Reviewer #2: Yes

2. Has the statistical analysis been performed appropriately and rigorously? 

Reviewer #1: Yes

Reviewer #2: Yes

3. Have the authors made all data underlying the findings in their manuscript fully available?

Reviewer #1: Yes

Reviewer #2: No

4. Is the manuscript presented in an intelligible fashion and written in standard English?

Reviewer #1: Yes

Reviewer #2: Yes

5. Review Comments to the Author

Reviewer #1: The authors here present data from an impressively large number of patients who underwent resection of liver metastases for their colorectal cancer. They examine the role of 5 biomarkers and conclude elevated expression of a combination of these biomarkers associate with worse survival outcomes. These findings are consistent with prior efforts and fit into the general understanding of clinical management of metastatic colorectal cancer. I think that the message could be strengthened with additional points to be addressed:

1. None of the markers for systemic inflammation are associated with clinical outcomes as a single marker in the multivariate analysis, and the significance is seen only when analyzed "in combination." It is unclear which markers, when "2 or more" were elevated, were actually elevated. If one is to translate these to the clinic, should he or she order all five, or were there specific ones which seemed to be more associated with outcome? As presented, the applicability here is very difficult for the reader.

- We thank the Reviewer for this comment. We have now explored this further by studying three groups of patients in an additional explorative analysis, namely patients with zero elevated markers, patients with CEA and/or CA19-9 elevated, and patients with YKL-40, IL-6 and/or CRP elevated. The results of this analysis are included as S6 Table in the Supporting Information files. They showed that patients with elevated CEA and/or CA19-9 elevated postoperatively had a significantly higher risk of relapse or death, while elevated YKL-40, IL-6, and/or CRP was not significantly associated with increased risk of relapse or death in any instance. We have already analyzed the biomarkers together in a multivariable analysis, although on a continuous scale, which showed that mainly CEA and CA19-9 were the drivers behind the negative prognostic value of having several elevated biomarkers, although YKL-40 was also associated with a negative prognostic outcome as a preoperative marker in relation to OS. 

We still conclude that the risk of relapse and death associated with an increase in the number of elevated markers, although unspecific, could be due to an additive effect of the markers, each highlighting a different aspect of disease relapse or imminent death. The sections “Results” and “Discussion” have been updated with comments on these explorative findings.

2. The authors mention that these findings support the use of these markers as a predictive biomarker and "might identify patients who benefit from more adjuvant chemotherapy." However, while there analyses may support the combination of markers PROGNOSTICALLY, the authors provide zero detail of receipt of any adjuvant chemotherapy for these patients. Therefore these claims are unsupported and should be removed as presented from the manuscript.

- We thank the Reviewer for this accurate observation. The manuscript text has been modified as follows (section “Abstract”):

“The results suggest that a protein panel of the inflammatory biomarkers YKL-40, IL-6, and CRP, and the cancer biomarkers CEA and CA19-9 might identify patients that benefit from more aggressive treatment and surveillance.”

3. One could argue that if this combination of markers is associated with a poor prognosis, the biology of the given CRC tumor is especially poor, and these might identify the patients who undergo resection that will have a higher risk of relapse and therefore NOT benefit from more surgery if at especially high risk for relapse and shortened OS. I disagree that re-resection would be warranted based on the data presented.

- We fully agree with the Reviewer. We have not included the information on possible re-resections in our data, and thus, their role cannot be evaluated based on our findings. Therefore, we have removed the claim in question from the manuscript and edited the text as follows: 

Section “Abstract”: “The results suggest that a protein panel of the inflammatory biomarkers YKL-40, IL-6, and CRP, and the cancer biomarkers CEA and CA19-9 might identify patients that benefit from more aggressive treatment and surveillance.”

Section “Conclusions”: “In the future, this panel might be used to select the patients that could benefit from more aggressive perioperative chemotherapy and follow-up.”

4. If IL-6 is responsible for producing CRP and YKL-40, then if IL-6 is elevated, should the other downstream markers be as well? If so, what is the reason for needed to test all three? IL-6 alone did not identify high-risk patients in the multivariate survival analysis. The authors could consider addressing this in their discussion.

- We thank the reviewer for this thoughtful question. It is true that IL-6 primarily stimulates the production of CRP in the liver, but YKL-40 is also stimulated by other biological processes. Since the studied cohort has metastases to the liver, we must assume some dysfunction of the hepatic production of CRP. This makes studying all parameters in this particular cohort interesting. We have updated the section “Discussion” addressing this aspect in a more elaborate manner.

5. Clarification of "major" and "minor" hepatic resections (how are these defined?) is needed in the methods section.

- We have defined the meaning of “minor” and “major” 1) in Materials and Methods, in the section “Patients” (“The liver resections were considered major if more than two Couinaud segments were resected.”), and 2) in Table 1 (“2 1–2 (minor) or ≥3 (major) Couinaud segments resected”).

6. YKL-40, IL-6 both increased from the preoperative to postoperative setting. One would expect that if these were associated with cancer only, there would be some decrease in median levels following resection, unless most patients recurred in the time before the first postoperative blood specimen was drawn. How do the authors interpret the fact that %patients with elevated inflammatory markers rose after surgery? Could these nonspecific markers be reflective of other factors?

- We thank the Reviewer for this interesting reflection. It is possible that some of the increase is explained by the surgical trauma, although the publication that we have as a reference in the Discussion suggests that this increase wanes after approximately 3 weeks. That is why we attribute – at least in part – the increase to micro-metastatic disease, although we acknowledge that this is partially only speculative.

We have now addressed the Reviewer’s questions in the section “Discussion” as follows: 

“In addition, serum YKL-40 has been shown to be increased until the 21st postoperative day merely due to the surgical trauma related to primary tumor resection [34]. We sampled blood a median of 94 (IQR 89–98) days after surgery. Thus, the postoperative increase seen in serum YKL-40 and IL-6 in some of the patients could be caused by micro-metastatic disease rather than the large surgical trauma in the liver, although it is possible that some effect of the surgical trauma persisted. We assume that the postoperative changes in inflammatory biomarkers would have normalized before the postoperative sampling.”

7. Without knowing which markers were elevated (2 of 5, 3 of 5 -which ones???), the clinical applicability of these findings is low, especially since, as the authors mention, these markers can be attributed to other cancers, autoimmune diseases, etc. The cost effectiveness of this approach is unclear, and the omission of alternate emerging approaches as a prognostic/predictive biomarker such as ctDNA in this setting by the authors is glaring. I am not sure the conclusions from the authors are supported by their data.

- We thank the Reviewer for this comment. We have now defined, which biomarkers were elevated and thus, which combinations were prognostic.

In our data, all the patients had only colorectal cancer, and other malignancies were an exclusion criterion. We have now mentioned this in the section “Patients” (“All patients included were diagnosed with resectable CRLM, and those with possible other malignancies were excluded.”). Our results suggest that the biomarkers are prognostic in colorectal cancer, but we cannot generalize this to apply to other cancers.

We have not included the information concerning possible autoimmune diseases in the database, because there were only a few – if any – patients with those diseases. We assume that if the increase in the biomarker levels would be related to autoimmune diseases instead of colorectal cancer, the number of patients with those diseases should be considerable to confound the biomarker-mortality association.

Concerning ctDNA, its role has by far been equivocal, and further research is needed. As the Reviewer suggests, it would have been interesting to investigate ctDNA in our material, but unfortunately, our samples do not fulfill the technical criteria for these analyses. We have now updated the section “Discussion” including ctDNA as an alternate approach in this field.

Reviewer #2: I would like the commend the authors on a extremely well written manuscript which seeks to evaluate whether elevated serum biomarker levels of YKL-40, IL-6, and CRP may improve upon prognostic significance beyond CEA and CA 19-9. The authors perform a retrospective evaluation of biomarker study of 441 consecutive patients undergoing resection for colorectal liver metastases, where the collected samples prospectively preoperatively and postoperatively. With a median follow-up of 5+ years, the authors found that patients with 2-5 biomarkers elevated were at increased risk of relapse and worse OS compared to those with 0-1 biomarkers elevated. This is a large cohort of patients with a well defined population of patients, with potential for important prognostic and therapeutic implications of treatment. I would ask the authors to clarify the following points to further strengthen their manuscript:

1) Given the awareness and prognostic significance of mutational status, as the authors acknowledged, this manuscript would be further improved if mutational status were available to correlate with inflammatory biomarkers. Was microsatellite status available on any of these patients?

- We fully agree with the Reviewer. Unfortunately, the microsatellite status was not routinely analyzed at our hospital prior to 2018. We have now specified this in the manuscript text as follows (section “Discussion”): 

“Information about the KRAS, NRAS and BRAF mutational status as well as the microsatellite instability (MSI) status could have been valuable, since the mutational status has been shown to predict recurrence patterns after liver resection [42, 43], but the mutational status was not routinely analyzed at Helsinki University Hospital prior to 2013 and the MSI not prior to 2018.”

2) From a statistical standpoint, was Bonferroni correction applied to adjust for multiple testing, given the authors evaluation of outcomes categorized as: 0 elevated/1 elevated/ 2 elevated/3 elevated/ 4 elevated/5 elevated and then subsequently as: 0-1 elevated/2-5 elevated. Also, were other categorizations evaluated as well (i.e 0-2 elevated/3-5 elevated) and if so, it further strengthens the role for application of Bonferroni correction.

- Bonferroni correction was not taken into account, and thus the results may be due to chance. The large number of patients included in the material increases the statistical validity of the results, but as stated in our Discussion, we acknowledge the statistical uncertainty surrounding our findings and state that our results will need external validation, whereby an independent cohort can show the same results in order to attach certainty to the results we obtain. We have tried to investigate our cohort using different approaches. The only comparison of number of elevated markers was 0–1 vs. 2–5, which can be seen as an exploration of testing each number of elevated markers against zero elevated markers. No other combinations were tested, which is why the use of Bonferroni correction is not justified by this particular aspect. We have updated the sections “Results” and “Discussion” mentioning the additional testing of 0–1 vs. 2–5 elevated markers, thus attaching less certainty to this finding, and we hope that this will alleviate the Reviewer’s concern.

3) How was 3 months postop chosen as the optimal time point for this analysis? Was there subsequent serial draws at other time points as part of the study as well?

- We thank the Reviewer for this question. We chose that time point, because we hypothesized that 3 months after liver resection, the possible postoperative complications would have resolved and the liver tissue would have regenerated, and thus, the surgical trauma would not affect the tumor marker levels anymore. We also estimated that earlier increase in the biomarker levels would most probably be due to either residual disease or surgical trauma and not recurrence. 

Serum samples were also drawn one week after resection, but those samples were not included in this study. Their prognostic value would have been controversial, since the levels of the inflammatory biomarkers rise postoperatively merely due to surgical trauma (Shantha Kumara HM et al. World J Gastrointest Oncol. 2016;8(8):607-14.).

4) Given increasing data evaluating early detection of disease recurrence with liquid biopsies, the authors should mention it as emerging technologies that are challenging the prognostic significance of traditional biomarkers like CEA/CA 19-9.

- We thank the Reviewer for this observation. We have now mentioned this in the section “Introduction” as follows: 

“The early detection of cancer with emerging technologies, such as liquid biopsies that consist of circulating tumor cells, DNA, and exosomes in the peripheral blood, is under active investigation [22]. However, protein biomarkers are at present an integral part of clinical work and may be more easily accessed.”

5) Interesting that CA 19-9 was the only significant biomarker on MVA in the preoperative setting for RFS and only CEA and CA 19-9 were significant in the postoperative setting on MVA. This suggests that perhaps CEA and CA 19-9 (known prognostic factors) are what is driving the significance of the 2-5 biomarkers, and not YKL-40, IL-6 and CRP. Where any of the other clinical variables on MVA significant?

- To respond to the Reviewer’s concerns in this aspect, we have explored this further by studying three groups of patients in an additional explorative analysis, namely patients with zero elevated markers, patients with elevated CEA and/or CA19-9, and patients with elevated YKL-40, IL-6 and/or CRP. The results of this analysis are included as the S6 Table in the Supporting Information files. They showed that patients with elevated CEA and/or CA19-9 postoperatively had a significantly higher risk of relapse or death, while elevated YKL-40, IL-6, and/or CRP was not significantly associated with increased risk of relapse or death in any instance. We have already analyzed the biomarkers together in a multivariate analysis, although on a continuous scale, which showed that mainly CEA and CA19-9 were the drivers behind the negative prognostic value of having several elevated biomarkers, although YKL-40 was also associated with a negative prognostic outcome as a preoperative marker in relation to OS. The sections “Results” and “Discussion” have been updated with comments on these explorative findings.

6) Table 1 as current presented is confusing and hard to follow. I would suggest that the authors remove outcome data from the Table 1 (alive/dead), and then present the biomarker data in the more concise and readable manner. Also please add R0/R1/R2 rates to Table. 1

- We have now edited the Table 1 according to the Reviewer’s suggestions. To present the biomarker data in a more readable manner, we separated them from the patient characteristics and created a new Table 2.

However, we decided to keep the outcome data included in Table 1, since we find that the information helps the readers to get a better overall understanding of the patient material.

7) On page 10, line 201: Please edit to include "WERE".....Ca 19-9 (HR 1.08) WERE associated

- We have corrected the sentence according to the Reviewer’s suggestion.

8) Similalry, on page 12: Please edit to include "WERE"....CA 19-9 (HR 1.12) WERE associated...

- We have corrected the sentence according to the Reviewer’s suggestion.

---

## [Decision Letter · Decision Letter 1]

29 Apr 2020

PONE-D-19-31995R1

Elevated serum YKL-40, IL-6, CRP, CEA, and CA19-9 combined as a prognostic biomarker panel after resection of colorectal liver metastases

PLOS ONE

Dear Dr. Peltonen,

Thank you for submitting your manuscript to PLOS ONE. After careful consideration, we feel that it has merit but does not fully meet PLOS ONE’s publication criteria as it currently stands. Therefore, we invite you to submit a revised version of the manuscript that addresses the points raised during the review process.

We would appreciate receiving your revised manuscript by Jun 13 2020 11:59PM. To enhance the reproducibility of your results, we recommend that if applicable you deposit your laboratory protocols in protocols.io, where a protocol can be assigned its own identifier (DOI) such that it can be cited independently in the future. For instructions see: http://journals.plos.org/plosone/s/submission-guidelines#loc-laboratory-protocols

We look forward to receiving your revised manuscript.

Kind regards,

Eugene J. Koay, M.D., Ph.D.

Academic Editor

PLOS ONE

Additional Editor Comments (if provided):

The reviewers have identified some concerns about the statistical approach. Please address these along with their other concerns if you wish to revise the manuscript again.

Reviewers' comments:

Reviewer's Responses to Questions

**Comments to the Author**

1. If the authors have adequately addressed your comments raised in a previous round of review and you feel that this manuscript is now acceptable for publication, you may indicate that here to bypass the “Comments to the Author” section, enter your conflict of interest statement in the “Confidential to Editor” section, and submit your "Accept" recommendation.

Reviewer #1: All comments have been addressed

Reviewer #2: (No Response)

Reviewer #3: (No Response)

2. Is the manuscript technically sound, and do the data support the conclusions?

Reviewer #1: Yes

Reviewer #2: Partly

Reviewer #3: Partly

3. Has the statistical analysis been performed appropriately and rigorously? 

Reviewer #1: Yes

Reviewer #2: I Don't Know

Reviewer #3: No

4. Have the authors made all data underlying the findings in their manuscript fully available?

Reviewer #1: Yes

Reviewer #2: Yes

Reviewer #3: Yes

5. Is the manuscript presented in an intelligible fashion and written in standard English?

Reviewer #1: Yes

Reviewer #2: Yes

Reviewer #3: Yes

6. Review Comments to the Author

Reviewer #1: The authors have addressed all comments with the submitted revision, and limitations to the generalizability to the findings have been adequately detailed in the Discussion section.

Reviewer #2: Overall, I appreciate the reviewer's responses to commentary.

The one persistent issue I would raise is why the authors place emphasis and suggest benefit of IL-6 and CRP when I am not sure the data shows this. When you tease out the results, it appears that CEA and CA 19-9 are driving the prognostic significance. CRP is not significant on any UVA or MVA either in preop or postop. IL-6 and YLK-40 are significant on UVA, but not MVA. And the authors even perform a requested analysis at reviewers' request that demonstrates: "We have now explored this further by studying three groups of patients in an additional explorative analysis, namely patients with zero elevated markers, patients with CEA and/or CA19-9 elevated, and patients with YKL-40, IL-6 and/or CRP elevated. The results of this analysis are included as S6

Table in the Supporting Information files. They showed that patients with elevated CEA

and/or CA19-9 elevated postoperatively had a significantly higher risk of relapse or

death, while elevated YKL-40, IL-6, and/or CRP was not significantly associated with

increased risk of relapse or death in any instance."

All in all, I somehow have a hard time wrapping my head around the conclusion that the "combined prognostic value of all 5 markers has benefit" as the results of the study do not support these findings.

Reviewer #3: The manuscript entitled ‘Elevated serum YKL-40, IL-6, CRP, CEA, and CA19-9 combined as a prognostic biomarker panel after resection of colorectal liver metastases’ with the aim to examine their associations with relapse-free survival and overall survival in combination with serum C-reactive protein (CRP), carcinoembryonic antigen (CEA), and carbohydrate antigen 19-9 (CA19-9) in patients with colorectal liver metastases.

This is an interesting study, however, the manuscript requires further improvement.

Comments

Abstract, all presentation of CI95% to be written as 95% CI. This apply throughout the manuscript i.e Page 10 Line 195, Page 24 Table 3, Page 27 Table 4, Supplementary Table 5A, 5B, 5C and 5D, 6

The abstract to be labelled with heading Background, Methods, Results, Conclusion.

Materials and methods

It would be good to include a study flowchart.

There was no sample size calculation or power of study from the sample size used was discussed in the manuscript.

Page 7 Line 151. P95% (subscript 95%) to be written as P95%.

Statistical analyses

Page 7 Line 155-158, the sentence on the function of statistical tests (Wilcoxon signed-rank test and Spearman’s rank correlation coefficient) incomplete and requires improvement.

Page 7 Line 163, the reason to use LOD/2 as one of the imputation methods for values below the LOD to be stated.

Page 7 Line 173, full name for AUC to be stated.

It would be good to include a brief description on the variables coding which were used in the analysis.

Results

Page 21 Table 1, the title is too short. Decimal points for percentages to be standardized.

Page 21 Table 1 and Page 22 Table 2 and text, the word range to be replaced or to be denoted as min-max.

Page 22 Table 2, symbol N to be replaced with n. N to be used for overall/Total sample size. Total sample size for each marker at pre and post operative to be stated.

Page 10 Line 202, 205, 205, 207, 210 the word pre operative or post operatively to be stated for YKL-40, CEA, CEA, CA19-9, CEA.

Page 24 Table 3, Page 27 Table 4, the 95%CI for the 5 elevated are wide. Perhaps data for 4 elevated and 5 elevated could be merged as one.

Page 25 & 26 Fig 1 & 2 and Fig 3 & 4 title, the number elevated to be stated in order to differentiate the titles.

Supplementary Table 5A-5D & Table 6, (0) to be omitted from Synchronous (0).

Model fit information for the multivariate analysis to be stated.

The statement/reason on 'Bonferroni correction' to be clearly stated in the discussion.

References did not conform to the journal format.

7. PLOS authors have the option to publish the peer review history of their article (what does this mean?). If published, this will include your full peer review and any attached files.

Reviewer #1: No

Reviewer #2: No

Reviewer #3: No

---

## [Author Response · Author response to Decision Letter 1]

1 Jun 2020

Responses to the comments of the Academic Editor and the Reviewers

We thank the Reviewers for accurate comments and constructive criticism. We have revised the manuscript according to the suggestions, and responses to the comments and questions point by point are shown below. We strongly feel that these changes have improved the manuscript.

Additional Editor Comments (if provided):

The reviewers have identified some concerns about the statistical approach. Please address these along with their other concerns if you wish to revise the manuscript again.

Reviewers' comments:

Reviewer's Responses to Questions

Comments to the Author

1. If the authors have adequately addressed your comments raised in a previous round of review and you feel that this manuscript is now acceptable for publication, you may indicate that here to bypass the “Comments to the Author” section, enter your conflict of interest statement in the “Confidential to Editor” section, and submit your "Accept" recommendation.

Reviewer #1: All comments have been addressed

Reviewer #2: (No Response)

Reviewer #3: (No Response)

2. Is the manuscript technically sound, and do the data support the conclusions?

Reviewer #1: Yes

Reviewer #2: Partly

Reviewer #3: Partly

3. Has the statistical analysis been performed appropriately and rigorously? 

Reviewer #1: Yes

Reviewer #2: I Don't Know

Reviewer #3: No

4. Have the authors made all data underlying the findings in their manuscript fully available?

Reviewer #1: Yes

Reviewer #2: Yes

Reviewer #3: Yes

5. Is the manuscript presented in an intelligible fashion and written in standard English?

Reviewer #1: Yes

Reviewer #2: Yes

Reviewer #3: Yes

6. Review Comments to the Author

Reviewer #1: The authors have addressed all comments with the submitted revision, and limitations to the generalizability to the findings have been adequately detailed in the Discussion section.

Reviewer #2: Overall, I appreciate the reviewer's responses to commentary.

The one persistent issue I would raise is why the authors place emphasis and suggest benefit of IL-6 and CRP when I am not sure the data shows this. When you tease out the results, it appears that CEA and CA 19-9 are driving the prognostic significance. CRP is not significant on any UVA or MVA either in preop or postop. IL-6 and YLK-40 are significant on UVA, but not MVA. And the authors even perform a requested analysis at reviewers' request that demonstrates: "We have now explored this further by studying three groups of patients in an additional explorative analysis, namely patients with zero elevated markers, patients with CEA and/or CA19-9 elevated, and patients with YKL-40, IL-6 and/or CRP elevated. The results of this analysis are included as S6 Table in the Supporting Information files. They showed that patients with elevated CEA and/or CA19-9 elevated postoperatively had a significantly higher risk of relapse or death, while elevated YKL-40, IL-6, and/or CRP was not significantly associated with increased risk of relapse or death in any instance."

All in all, I somehow have a hard time wrapping my head around the conclusion that the "combined prognostic value of all 5 markers has benefit" as the results of the study do not support these findings.

- We thank the Reviewer for again taking the time to review our work in this second revision. As for the above-mentioned concern, we agree to some extent with the Reviewer, and we are glad that the other changes made in the latest revision were of satisfactory quality. We have addressed the Reviewer’s concern by changing our wording to be less reflective of a perceived benefit of our biomarker panel as it, indeed, cannot be said that it unequivocally has benefit (please see lines 57–58 in the abstract, lines 357–359, and line 378 in the present revised manuscript).

Reviewer #3: The manuscript entitled ‘Elevated serum YKL-40, IL-6, CRP, CEA, and CA19-9 combined as a prognostic biomarker panel after resection of colorectal liver metastases’ with the aim to examine their associations with relapse-free survival and overall survival in combination with serum C-reactive protein (CRP), carcinoembryonic antigen (CEA), and carbohydrate antigen 19-9 (CA19-9) in patients with colorectal liver metastases.

This is an interesting study, however, the manuscript requires further improvement.

Comments

Abstract, all presentation of CI95% to be written as 95% CI. This apply throughout the manuscript i.e Page 10 Line 195, Page 24 Table 3, Page 27 Table 4, Supplementary Table 5A, 5B, 5C and 5D, 6.

- We have made the corrections according to the Reviewer’s suggestion.

The abstract to be labelled with heading Background, Methods, Results, Conclusion.

- We have corrected the Abstract according to the Reviewer’s suggestion.

Materials and methods

It would be good to include a study flowchart.

- We have now included in the manuscript a study flow diagram as a Supplementary Figure 1.

There was no sample size calculation or power of study from the sample size used was discussed in the manuscript.

- We did not perform a post-hoc power calculation nor do we find it necessary. Most post-hoc power estimations can be easily biased. As the present study was not planned prior to the collection of samples, an a priori sample size calculation was not needed for this specific study. Power estimations are essential when applying for ethics approval, but this is done at a much earlier stage in planning.

Page 7 Line 151. P95% (subscript 95%) to be written as P95%.

- The correction is made according to the Reviewer’s suggestion.

Statistical analyses

Page 7 Line 155-158, the sentence on the function of statistical tests (Wilcoxon signed-rank test and Spearman’s rank correlation coefficient) incomplete and requires improvement.

- We thank the Reviewer for bringing our attention to this issue, although we do not appreciate in the same way what is incomplete about the sentence. We explicitly state that these tests were used to test for pre- and postoperative differences in serum values, and we use a Spearman’s rank test for testing for correlations. Later in the manuscript we mention the results of correlations and p-values for differences in the serum values pre- and postoperatively. If the Reviewer continues to find the sentence erroneous, please direct us to what is specifically incomplete about the sentence and the specific improvements needed.

Page 7 Line 163, the reason to use LOD/2 as one of the imputation methods for values below the LOD to be stated.

- The reason is that we believed the most correct value was a non-zero value, and chose a pragmatic approach to dealing with “missing” values. Please see the lines 165–167.

Page 7 Line 173, full name for AUC to be stated.

- The correction is made according to the Reviewer’s suggestion.

It would be good to include a brief description on the variables coding which were used in the analysis.

- We only coded the variable number of elevated markers by simply counting the number of elevated markers a patient had according to the already mentioned cut points. We specifically state the investigated cut points in the Methods section, and we hope that this is sufficient for the Reviewer.

Results

Page 21 Table 1, the title is too short. Decimal points for percentages to be standardized.

- We have now standardized the percentages and lengthened the title.

Page 21 Table 1 and Page 22 Table 2 and text, the word range to be replaced or to be denoted as min-max.

- The word “range” is now replaced by “min.–max.”.

Page 22 Table 2, symbol N to be replaced with n. N to be used for overall/Total sample size. Total sample size for each marker at pre and post operative to be stated.

- The corrections are made according to the Reviewer’s suggestion.

Page 10 Line 202, 205, 205, 207, 210 the word pre operative or post operatively to be stated for YKL-40, CEA, CEA, CA19-9, CEA.

- The section is now edited according to the Reviewer’s suggestion.

Page 24 Table 3, Page 27 Table 4, the 95%CI for the 5 elevated are wide. Perhaps data for 4 elevated and 5 elevated could be merged as one.

- These confidence intervals are wide since very few patients have all five markers elevated. We do not believe that rearranging groups according to the confidence intervals they present in the analyses is a valid method of defining groups, and it could easily lead to statistical cosmetic improvement. We hope that the readers will interpret the confidence intervals warily and accordingly, and we ourselves do not put emphasis in these findings.

Page 25 & 26 Fig 1 & 2 and Fig 3 & 4 title, the number elevated to be stated in order to differentiate the titles.

- The corrections are made according to the Reviewer’s suggestion.

Supplementary Table 5A-5D & Table 6, (0) to be omitted from Synchronous (0).

- The corrections are made according to the Reviewer’s suggestion.

Model fit information for the multivariate analysis to be stated.

- We state in the section “Methods” that the “Proportional hazards assumption was evaluated by testing for trends in the scaled Schoenfeld residuals” (lines 174–175). This should suffice for fitting of the models as no consensus exists on the correct model fit number to compare Cox regression models.

The statement/reason on 'Bonferroni correction' to be clearly stated in the discussion.

- We have now edited the section “Discussion” according to the suggestion (lines 364–365).

References did not conform to the journal format.

- We use the EndNote style format provided by the journal itself. If any issues should arise later in the editorial process, we trust it that the editorial staff will correct any mistakes in typesetting and direct us to the correct format of references.

7. PLOS authors have the option to publish the peer review history of their article. If published, this will include your full peer review and any attached files.

Do you want your identity to be public for this peer review? For information about this choice, including consent withdrawal, please see our Privacy Policy.

Reviewer #1: No

Reviewer #2: No

Reviewer #3: No

---

## [Decision Letter · Decision Letter 2]

3 Jul 2020

PONE-D-19-31995R2

Elevated serum YKL-40, IL-6, CRP, CEA, and CA19-9 combined as a prognostic biomarker panel after resection of colorectal liver metastases

PLOS ONE

Dear Dr. Peltonen,

Thank you for submitting your manuscript to PLOS ONE. After careful consideration, we feel that it has merit but does not fully meet PLOS ONE’s publication criteria as it currently stands. Therefore, we invite you to submit a revised version of the manuscript that addresses the points raised during the review process.

The manuscript is acceptable with one minor revision in the Discussion as requested by Reviewer 1. Please address this point in the Discussion and resubmit.

We look forward to receiving your revised manuscript.

Kind regards,

Eugene J. Koay, M.D., Ph.D.

Academic Editor

PLOS ONE

Reviewers' comments:

Reviewer's Responses to Questions

**Comments to the Author**

1. If the authors have adequately addressed your comments raised in a previous round of review and you feel that this manuscript is now acceptable for publication, you may indicate that here to bypass the “Comments to the Author” section, enter your conflict of interest statement in the “Confidential to Editor” section, and submit your "Accept" recommendation.

Reviewer #1: All comments have been addressed

Reviewer #2: All comments have been addressed

Reviewer #3: All comments have been addressed

2. Is the manuscript technically sound, and do the data support the conclusions?

Reviewer #1: Partly

Reviewer #2: Yes

Reviewer #3: Partly

3. Has the statistical analysis been performed appropriately and rigorously? 

Reviewer #1: Yes

Reviewer #2: Yes

Reviewer #3: Yes

4. Have the authors made all data underlying the findings in their manuscript fully available?

Reviewer #1: Yes

Reviewer #2: Yes

Reviewer #3: Yes

5. Is the manuscript presented in an intelligible fashion and written in standard English?

Reviewer #1: Yes

Reviewer #2: Yes

Reviewer #3: Yes

6. Review Comments to the Author

Reviewer #1: I think that the authors have made their case about the five biomarkers here. It still remains very unclear the clinical practicality of these findings, as I could not envision these ultimately making it to the clinic for routine use, especially in the emerging era of ctDNA technologies, which are more sensitive and specific than the biomarker combinations detailed here. I think that acknowledgement of the ctDNA data for the metastatic CRC resections as a prognostic biomarker could help further put the authors' conclusions into a more accurate context. I think it is more likely that ctDNA would be the more feasible route for personalizing management of perioperative therapies. I would recommend the authors updating the discussion to include some of the ctDNA findings.

Reviewer #2: Thank you very much for the responses to my concerns. The authors have adequately addressed my concerns.

Reviewer #3: (No Response)

7. PLOS authors have the option to publish the peer review history of their article (what does this mean?). If published, this will include your full peer review and any attached files.

Reviewer #1: No

Reviewer #2: No

Reviewer #3: No

---

## [Author Response · Author response to Decision Letter 2]

6 Jul 2020

Responses to the comments of the Academic Editor and the Reviewers

We thank the Reviewer #1 for the accurate comment and constructive criticism. We have revised the manuscript according to the suggestion, and our response is shown below.

Additional Editor Comments (if provided):

The manuscript is acceptable with one minor revision in the Discussion as requested by Reviewer 1. Please address this point in the Discussion and resubmit.

Reviewers' comments:

6. Review Comments to the Author

Reviewer #1: I think that the authors have made their case about the five biomarkers here. It still remains very unclear the clinical practicality of these findings, as I could not envision these ultimately making it to the clinic for routine use, especially in the emerging era of ctDNA technologies, which are more sensitive and specific than the biomarker combinations detailed here. I think that acknowledgement of the ctDNA data for the metastatic CRC resections as a prognostic biomarker could help further put the authors' conclusions into a more accurate context. I think it is more likely that ctDNA would be the more feasible route for personalizing management of perioperative therapies. I would recommend the authors updating the discussion to include some of the ctDNA findings. 

- We thank the Reviewer for this observation. We have now edited the section "Discussion" and pondered the prognostic value of the biomarkers investigated in this study in relation to the new emerging technologies, such as liquid biopsies, as follows (lines 361–369):

"It is probable that new technologies, liquid biopsies in particular, will be applied to clinical use quite shortly. They have potential in the detection and management of CRC, as they offer a non-invasive method for early detection of cancer, prognostic and predictive information, monitoring of treatment response, and identification of minimal residual disease [41]. In conjunction with resection of CRLM, elevated pre- [42] and postoperative [43] ctDNA levels have shown high concordance with relapse in ctDNA-positive patients. However, it seems unlikely that liquid biopsies could provide information on inflammatory responses, which affect the prognosis of CRC. Thus, the inflammatory biomarkers have an important role also in this era of advances in these new technologies."

41. Yamada T, Matsuda A, Koizumi M, Shinji S, Takahashi G, Iwai T, et al. Liquid Biopsy for the Management of Patients with Colorectal Cancer. Digestion. 2019;99(1):39-45. doi: 10.1159/000494411.

42. He Y, Ma X, Chen K, Liu F, Cai S, Han-Zhang H, et al. Perioperative Circulating Tumor DNA in Colorectal Liver Metastases: Concordance with Metastatic Tissue and Predictive Value for Tumor Burden and Prognosis. Cancer Manag Res. 2020;12:1621-30. doi: 10.2147/CMAR.S240869. PubMed PMID: 32184665.

43. Benešová L, Hálková T, Ptáčková R, Semyakina A, Menclová K, Pudil J, et al. Significance of postoperative follow-up of patients with metastatic colorectal cancer using circulating tumor DNA. World J Gastroenterol. 2019;25(48):6939-48. doi: 10.3748/wjg.v25.i48.6939. PubMed PMID: 31908397.

Reviewer #2: Thank you very much for the responses to my concerns. The authors have adequately addressed my concerns.

Reviewer #3: (No Response)

---

## [Editor Report · Decision Letter 3]

10 Jul 2020

Elevated serum YKL-40, IL-6, CRP, CEA, and CA19-9 combined as a prognostic biomarker panel after resection of colorectal liver metastases

PONE-D-19-31995R3

Dear Dr. Peltonen,

We’re pleased to inform you that your manuscript has been judged scientifically suitable for publication and will be formally accepted for publication once it meets all outstanding technical requirements.

Kind regards,

Eugene J. Koay, M.D., Ph.D.

Academic Editor

PLOS ONE

Additional Editor Comments (optional):

The authors have adequately addressed the minor revision. Thank you.
---

## [Editor Report · Acceptance letter]

15 Jul 2020

PONE-D-19-31995R3 

Elevated serum YKL-40, IL-6, CRP, CEA, and CA19-9 combined as a prognostic biomarker panel after resection of colorectal liver metastases 

Dear Dr. Peltonen:

I'm pleased to inform you that your manuscript has been deemed suitable for publication in PLOS ONE. Congratulations! Your manuscript is now with our production department. 

Kind regards, 

on behalf of

Dr. Eugene J. Koay 

Academic Editor

PLOS ONE